# Rabies as a Public Health Concern in India—A Historical Perspective

**DOI:** 10.3390/tropicalmed5040162

**Published:** 2020-10-21

**Authors:** Sreejith Radhakrishnan, Abi Tamim Vanak, Pierre Nouvellet, Christl A. Donnelly

**Affiliations:** 1MRC Centre for Global Infectious Disease Analysis, Department of Infectious Disease Epidemiology, Imperial College London, London W2 1PG, UK; Pierre.Nouvellet@sussex.ac.uk (P.N.); christl.donnelly@stats.ox.ac.uk (C.A.D.); 2Department of Animal Husbandry, Government of Kerala, Thiruvananthapuram 695 033, India; 3Ashoka Trust for Research in Ecology and the Environment, Bengaluru 560 064, India; avanak@atree.org; 4DBT/Wellcome Trust India Alliance Fellow, Hyderabad 500 034, India; 5School of Life Sciences, University of KwaZulu-Natal, Durban 4001, South Africa; 6School of Life Sciences, University of Sussex, Brighton BN1 9QG, UK; 7Department of Statistics, University of Oxford, Oxford OX1 3LB, UK

**Keywords:** stray dogs, Pasteur Institute, vaccination, colonial, British India, Civil Veterinary Department

## Abstract

India bears the highest burden of global dog-mediated human rabies deaths. Despite this, rabies is not notifiable in India and continues to be underprioritised in public health discussions. This review examines the historical treatment of rabies in British India, a disease which has received relatively less attention in the literature on Indian medical history. Human and animal rabies was widespread in British India, and treatment of bite victims imposed a major financial burden on the colonial Government of India. It subsequently became a driver of Pasteurism in India and globally and a key component of British colonial scientific enterprise. Efforts to combat rabies led to the establishment of a wide network of research institutes in India and important breakthroughs in development of rabies vaccines. As a result of these efforts, rabies no longer posed a significant threat to the British, and it declined in administrative and public health priorities in India towards the end of colonial rule—a decline that has yet to be reversed in modern-day India. The review also highlights features of the administrative, scientific and societal approaches to dealing with this disease in British India that persist to this day.

## 1. Introduction

“A bite from a mad dog is more dreaded than anything I know; which arises from the horribleness of the disease, the uncertainty of the animal’s being mad, or of the infection being received: The not knowing at what period to expect the effects, or to feel confident of having escaped it, keeps the person in a state of cruel suspence (sic) for months, or even years.”—Daniel Johnson, Sketches of Field Sports as followed by The Natives of India with observations on the animals (1822).

The World Health Organization (WHO) identifies neglected tropical diseases (NTDs) as a group of communicable diseases affecting over a billion people globally, primarily those living in poverty in low- and middle-income countries in the tropics and subtropics [1], imposing a significant economic burden on these countries. India has the dubious distinction of bearing the largest burden of at least 11 of these NTDs [2]. This list includes rabies, a viral infection caused primarily by the bite of infected hosts belonging to the mammalian order Carnivora (although all mammals, and in exceptional circumstances birds, can be infected), and less frequently the deposition of saliva on wounds or mucous membranes. In addition, it has the highest mortality rate of all known infectious agents, with nearly all individuals who develop clinical symptoms eventually dying [3]. Rabies has been eliminated (or historically been absent) in Western Europe and several island nations such as Japan, Australia and New Zealand (except for imported cases). In North America, domestic animal rabies in dogs and cats occurs infrequently, mainly through exposure to infected wildlife reservoir hosts [4]. However, in most rabies-endemic countries in Africa and Asia including India, domestic dogs are the main rabies reservoir and source of human exposure [4,5].

Of the 59,000 annual human deaths estimated to occur globally due to dog-mediated rabies, about 35% occur in India [4]. Over three-quarters of cases in India occur in rural communities with poor access to diagnostic facilities and post-exposure prophylaxis, which are key to preventing development of disease [5]. More than 95% of cases are caused by dog bites, largely because of the approximately 60 million stray/free-ranging dogs in the country [6], and many cases of human rabies go undetected, are misdiagnosed or are under-reported [7]. A significant proportion of cases (over one-third in a recent study) are children, and despite the availability of safe and effective vaccines, awareness of and access to post-exposure prophylaxis (PEP), including rabies immunoglobulin, continue to be poor [8].

Despite the high burden of human rabies in India, the disease is not notifiable and a structured surveillance system is yet to be put in place [7]. Rabies is not included in the list of diseases for which surveillance is routinely carried out by states and reported under the Integrated Disease Surveillance Project of the Indian Ministry of Health and Family Welfare [9]. Instead, dog bites and snake bites are to be reported separately (Diseases under surveillance: Presumptive (P form)) [9]. The absence of an organised national or regional system for rabies surveillance compounds the problem of poor availability of human and animal rabies incidence data [10]. Current estimates of the burden of rabies in India (over 20,000 human deaths annually) are based on an epidemiological study conducted in 2003 [5], and even this may be an underestimate of the true disease burden. Another study estimated that 12,700 human deaths from symptomatically identifiable furious rabies occurred in India in 2005 [11]. Most recently, a multicentric survey conducted in 2017 across seven Indian states estimated an annual incidence of animal exposures (bite, scratch or lick) of 1.26%, which was reportedly lower than previous estimates from India [12]. However, the authors acknowledged that, owing to the limited scale of their study, results could not be used to generate a country-level burden of potential rabies exposures.

A perusal of the Five-Year Plans for national development in independent India (post-1947) covering the period 1951 (when the first five-year plan was unveiled) to 2002 reveals that rabies was never prioritised for control. During this period, the very term “rabies” appears only twice—once in the fourth plan (1969–1974) while listing the diseases for which research was conducted at the Central Research Institute at Kasauli and once in the sixth plan (1980–1985) in a brief description of mortality rates of environmentally linked diseases (“Diseases like TB, Gastro-intestinal infections, malaria, filaria, infectious hepatitis, rabbies (sic) and hook worm …”) [13]. The 10th plan (2002–2007) mentions the development of a new animal rabies vaccine “being tested for technology transfer”, as well as research projects on a number of infectious diseases including rabies, although no further details are provided.

It is only in the 11th plan (2007–2012) that rabies control efforts are first mentioned in the form of pilot projects for the control of human rabies, for which 8.65 crore rupees (~2.1 million US dollars at the time) were allocated. For the first time in a Five-Year Plan, rabies control in animals, animal birth control and vaccination of stray dogs are also mentioned in this plan, as components of animal welfare to be handled by the Animal Welfare Board of India [13].

In 2014, the Ministry of Health and Family Welfare of the Government of India announced funding for a National Rabies Control Programme as part of the 12th five-year plan (2012–2017) [14]. This programme is coordinated by the National Centre for Disease Control, New Delhi, and the Animal Welfare Board of India, with the aim of halving human rabies deaths by the end of 2017. However, little information is available on the achievements of the programme, which finds no mention on the website of the national Ministry of Health and Family Welfare (https://mohfw.gov.in/) or the NITI Aayog (https://niti.gov.in/), which replaced the Planning Commission of India in 2015. A search for the term “rabies” on the Open Government Data Platform India (https://data.gov.in) returns no results [15]. The annual budget for 2018 presented by the Finance Minister of India allocated 40 crore rupees (approximately 6.13 million US dollars) for a few pilot schemes under the National Rural Health Mission, which includes control of human rabies [16]. This amount has been reduced to 25 crore rupees (approximately 3.51 million US dollars) in 2019 [17] and 2020 [18].

Given this background, it is natural to assume that rabies has always been accorded low priority in India. However, a quick glance through the literature on rabies in pre-independence India (before 1947) suggests otherwise. Rabies was one of several “tropical” afflictions, including cholera, plague, typhoid, tuberculosis, polio and snakebites, that were viewed as serious medical and public health problems, particularly for British residents in India. Consequently, it was subjected to great research and control efforts by the British colonial Government of India (hereafter referred to as GoI), frequently using native Indians as subjects for experimentation to develop and refine vaccines [19]. The effort to combat rabies and other infectious diseases was instrumental in the establishment of a wide network of research institutes in India and some important breakthroughs in development of rabies vaccines. However, the disease appears to have gradually lost priority in scientific circles and the colonial GoI, which may be the basis for its continued neglect in modern India. In this historical context, underlying reasons for the present-day underprioritisation of rabies in post-independence India need to be explored, as these may provide insights into what needs to change in order for rabies control in India to receive the priority and resources it deserves.

We used the search terms “rabies” (and variations of its spelling—“rabbies”, “rabeis” and “rabes”), “hydrophobia” and “India” to review a range of historical archives and online and physical documents about rabies in pre-independence India (covering the period from the early 1800s to 1947, when India gained independence from British rule, and the few years immediately after). These included articles published in scientific journals (via PubMed and Google Scholar) and popular magazines, historical documents held at the India Office Records and Private Papers of the British Library and the Wellcome Library at the Wellcome Collection, online archives of the Medical History of British India maintained by the National Library of Scotland (https://digital.nls.uk/indiapapers/), British Parliamentary Papers available via ProQuest UK Parliamentary Papers, historical newspapers available via ProQuest Historical Newspapers and documents available online at the Hathi Trust (www.hathitrust.org) and libraries of the universities of Oxford and Cambridge.

## 2. Rabies Documentation in Pre-Independence India

As one of the oldest diseases known to man, rabies was widely documented by the earliest human civilisations [20]. A disease akin to rabies was recognised in ancient Indian treatises on health and medicine. The *Susruta Samhita* (*Susruta’s Compendium*) is an ancient Indian text of Ayurveda (written between 1000 BCE and the first or second century CE), the Indian system of traditional medicine still practised in most parts of the country. This text details various medical conditions and surgical procedures and discusses in detail the symptoms of rabies in humans bitten by rabid dogs or wildlife, recognising that once symptoms develop in human bite victims, the disease is inevitably fatal [21]. The Mughal emperor Jahangir (1569–1627) is recorded to have noted the symptoms of rabies in an elephant that he owned [22]. It is also highly likely that rabies was documented extensively in the numerous vernacular languages on the Indian subcontinent.

Accounts of British medical and military personnel who worked in India during the 1800s highlight the fact that rabies, also referred to as hydrophobia, was widespread throughout India, responsible for the deaths of numerous Indian, British and European citizens [23,24,25]. The disease also caused extensive mortality in livestock and pet animals such as purebred dogs owned by British officials [23,26]. These accounts identified the occurrence of large populations of free-roaming (“stray”) dogs and to a lesser extent wildlife, predominantly jackals, as the main source of infection [23,25,27]. A collection of observations on life in India by a former surgeon of the East India Company (1822) includes a chapter titled “Observations on hydrophobia and rabid animals” that describes symptoms in humans in graphic detail [23]. The same chapter and other reports provide detailed descriptions of the progress of rabies in infected pet dogs and wildlife, recounting behavioural changes as symptoms began to manifest [23,26,27]. These symptoms included changes in temperament with increased displays of affection or misdirected aggression, changes in vocalisation which were often noticed by Indian caretakers and changes in appetite, varying from voracious consumption of food to eventual rejection. In one instance, a rabid pup that was bitten by a (presumably rabid) hyena interrupted a dance party, resulting in the party having to be broken up and the pup being killed immediately [23].

These accounts also detail the experiences of British military doctors who often had to treat patients with symptoms of rabies and their agony at having to witness progression of the disease and inevitable death [23]. Much effort was put into discovering ways to treat infected individuals and potential modes of treatment, including traditional Indian cures, were keenly discussed in medical circles [24,28,29]. Even at this time, it was well recognised that treating bite wounds as soon as possible after bites occurred was key to preventing disease progression [23,24]. A letter to the editor *of The Lancet* in 1829 discusses the symptoms of rabies, disputing whether it should also be referred to as “hydrophobia”, and possible ways of treatment including bleeding of patients in India [29]. A booklet on Ayurvedic treatments for various illnesses published in 1876 from Cochin, in present-day Kerala, includes symptoms of rabies and traditional treatment methods for exposure to “peppatti visham” (poison from a rabid/mad dog) such as chants, and pills and powders made from plant parts [30]. Various other treatments including Buisson baths [31] and cauterising wounds with caustic agents (e.g., nitric acid) [24,26] have also been documented. Rabies was also a significant health concern for British military personnel stationed in India, and pensions were given to the family of military personnel who died of rabies contracted in the line of duty [32].

Various aspects of rabies also found their way into Indian and British newspapers and magazines, ranging from individual theories about how the disease occurs (“a disease engendered by the practice in England of docking the tails of so many of our sporting and household dogs.”) (1861) [33]; reports of incidents of rabid dog bites [34] and outbreaks in wildlife [35]; descriptions of encounters with rabid dogs, symptoms observed and suggested control measures (“lunar caustic … apply it well to every wound.”) (1859) [36]; and an account of a former army officer who claimed to have successfully recovered from rabies after being exposed in India with a detailed description of his symptoms (1836) [37]. Indian newspapers also reprinted articles about rabies that were initially published in British newspapers [38].

Such news reports and readers’ letters to editors make evident that stray dogs, dog bites and rabies were an important public concern, particularly in major cities like Bombay (present-day Mumbai), Poona (present-day Pune), Lahore and Calcutta (present-day Kolkata) where many British and European citizens lived [39,40,41,42,43,44] and where significant numbers of cases were often reported [45,46]. Public awareness about rabies among British residents would also have been high when rabies was a major threat in Britain during the Victorian era and for a long time after its elimination in 1902 [47]. Complaints about “mad dogs” in India can be found in letters published as early as 1861 [33]. In addition to humans, purebred pet dogs were frequently infected [39,43] and British residents constantly demanded action from authorities to control rabies and stray dog populations [39,48,49], even proposing that private contributions be used to fund control measures [44]. Such concerns about rabies control also need to be located within discourses of sanitation, hygiene and urban improvement that were emerging in British India since the late 1800s [50]. These discourses were a product of the burden imposed by infectious diseases on British army personnel in India [50], and in rapidly expanding cities like Bombay and Calcutta, where epidemics of plague, cholera, measles and smallpox were frequently reported, particularly among the city’s poor [51,52].

## 3. Pasteur Institutes and Rabies Vaccination in British India

The discovery of a rabies vaccine by Louis Pasteur, Emile Roux and other colleagues in 1885 [20,53] was a ground-breaking medical milestone, resulting in the establishment of Pasteur Institutes (PIs) in various parts of the world for production of rabies vaccines [54]. Initially, individuals exposed to rabies in India had to undertake a long journey to the PI in Paris for treatment, thereby affecting their chances of survival [54]; such journeys were often reported in newspapers [27,34,55,56,57]. These journeys were a major financial burden for the GoI, by one estimate costing £100 per person treated in 1911 (approximately £12,000 per person in 2019 terms) [58]. Recognising the need to bring rabies treatment to India (“if only for the protection of Europeans, and especially of the troops.”), AV Lingard, Imperial bacteriologist at the Imperial Bacteriological Laboratory at Poona proposed in 1891 that “anti-rabic treatment and cure” could be started in the laboratory [54]. There was a public movement in the 1890s in India to gather support for the establishment of such institutes, described in detail by Chakrabarti (2012) [19]. The first PI in India started functioning at the hill station of Kasauli in 1900 under David Semple, a medical officer of the colonial Indian Medical Service [59]. It has been argued that this shift in the choice of locations from hot and humid Pune to the colder environment of Kasauli was driven primarily by a desire to maintain a distance from the native Indian population and to avoid the hot tropical climate of the Indian plains, rather than by scientific considerations [19].

Within a short period, the PI at Kasauli served as the main destination for treating an increasing numbers of individuals, both civilians and soldiers, exposed to rabies using vaccine produced at the institute [60,61] providing significant financial savings to the GoI by avoiding the costs of travel to Paris for treatment [58]. As a result of political pressure to decentralise rabies vaccination [19], PIs (or Pasteur sections within other institutions) were established throughout British India including at Coonoor in South India (1907) [62], Rangoon (in present-day Myanmar) (1915) [63], Shillong (East India) (1917), Bombay (1922) [64], Calcutta (1924) [19] and Patna (1928) [65]. Patients who were exposed through bites would often seek medical advice by sending a telegram to the PI, before deciding on travelling to the institute for treatment [66]. These PIs served thousands of individuals exposed to rabies from all parts of British India and Ceylon (present-day Sri Lanka) [67], even after India gained independence [68,69] and many continue to serve the same function to the present day [70].

Detailed statistics were collected on bite victims presenting for treatment to record information about which species bit them (dog, jackal, etc.), location, number, category and severity of bites (bites on head or face, bites through clothing, etc.), whether they completed the full course of vaccinations and the number of deaths post-vaccination—information which greatly improved scientific understanding about rabies [71]. From 1912, statistics on the number of individuals bitten by rabid animals and not seeking treatment were also compiled at Kasauli [61,72]. Hundreds of animals were also examined every year at PIs, veterinary colleges and other institutions like the Haffkine Institute in Bombay [61,62,73] to confirm a diagnosis of rabies. Thousands of copies of a pamphlet titled “Rabies and antirabic treatment in India” were printed and sent to local governments, with suggestions to translate these into local languages [63]. Updated editions of this pamphlet were published in subsequent years [74,75]. At one point, the Kasauli institute treated more patients every year than any other PI around the world [61,76].

Kasauli also became the site for extensive research into safer and more effective rabies vaccines, since the vaccines in use at the time often resulted in serious neurological complications [77]. One of the key events in the history of rabies vaccines was the development of a phenol-inactivated nerve tissue vaccine by David Semple based on Pasteur’s original work and developed through experiments and trials on patients at Kasauli [78]. Used for decades in large parts of the world, production of the Semple vaccine has now been discontinued, although it is still produced for human or animal use in a few countries in Africa [4]. The development and evolution of these and other modern rabies vaccines have been covered in detail elsewhere [19,79,80].

Eventually, post-exposure treatment was also decentralised by opening “outcentres” throughout India, though not without opposition from John Cunningham, the Director of the PI at Kasauli in the 1920s who wanted to expand research on rabies vaccines at the institute [59,81]. A 1923 news report identifies such centres “at Karachi, Allahabad, Ahmednagar, Poona, Belgaum and Karwar” as well as Parel in Bombay [42]. These outcentres made it possible to greatly reduce delays in post-bite treatment, and the mortality rate among treated individuals in 1938 was reported to be 0.52%. By 1938, the Kasauli institute had over 140 outcentres in the northern provinces and other Indian states, while the Coonoor institute had 223 outcentres in Madras Presidency and southern states [82]. While public funds and government grants initially supported the establishment and functioning of PIs at Kasauli and Coonoor, the effectiveness of and demand for rabies vaccines developed at these centres meant that by the 1920s, these institutions started to function fully as private entities, with most of their income coming from the sale of rabies vaccines to government, municipal and local bodies and state hospitals [82].

## 4. Rabies Control in Animals

One of the earliest documented pieces of legislation for dealing with stray and rabid dogs in British India is regulation II of 1813, which permitted the destruction of ownerless dogs in Bombay city during specific periods of the year. The strict (and often overenthusiastic) enforcement of this regulation sometimes led to the destruction of owned dogs as well and is closely associated with what has been described as the “Bombay dog riots” of 1832. These riots, which also had communal overtones, have been described in detail elsewhere [83]. Other legislation included ‘section 68 of the Cantonment Code of 1912’, and provisions in Municipal Acts, which authorised cantonments or municipalities to detain or destroy confirmed or suspected rabid dogs as well as stray dogs. In municipalities, officers of the Civil Veterinary Department (CVD) were authorised to carry out these functions. Some Local Self-Government Acts also permitted issuing rewards for destruction of “noxious animals” [84].

A newspaper report from 1912 describes the system in Madras where dog capture was outsourced to “low caste dog-catchers”, and dogs were “painlessly destroyed in a lethal chamber” [85]. It was proposed that a similar system be implemented in Bombay. A news report from 1923 describes the efforts of the Health Department of Bombay municipality in “diminishing the number of dangerous, diseased and stray dogs in the city”. This was carried out by a team of “3 sub-inspectors, 2 dog carts, 8 cart drivers, 18 dog catchers and a lethal chamber … in which dogs are destroyed by means of carbonic acid gas”. The municipality reportedly spent about 10,000 rupees a year for this purpose, destroying 6579 and 6848 “ownerless dogs” and returning 22 and 6 dogs to their owners in 1921 and 1922, respectively [42]. Similar efforts were also reported from Calcutta [40] and Poona [41]. Lethal chambers and “electrocutors” for destruction of dogs were installed in local bodies—a report of the CVD of Madras Presidency (1929–30) describes the inspection of lethal chambers in Tiruppur, Coonoor and Pollachi, construction of additional chambers at Erode and Udamalpet and an “electrocutor” at Ootacamund (present-day Ooty) [73]. However, one letter from a reader describing empty dog carts and the number of dogs on the road [39] suggests that such measures may have been no more effective in controlling dog populations and rabies than they were in more contemporary times. These measures and the methods used to kill dogs (carbonic acid gas, strychnine poisoning, clubbing to death, electrocution) [19,86] were opposed by many Indians due to religious reasons [41,83], by Indian vernacular newspapers and many British residents [19]. In addition to dogs, wildlife [74], predominantly jackals [87] were also often destroyed.

A host of other measures targeting owned dogs were largely modelled on measures implemented in Britain in the 1800s that had proven successful in making the country rabies-free by 1902 [47]. Officials recognised that owned dogs in India were often unconfined and thus could be infected with or spread rabies—one report proposed that owned female dogs that were allowed to roam freely when in oestrus resulted in increased dog fights and thus the spread of rabies [88]. Purebred dogs were often allowed to roam freely [89] or used by European soldiers to hunt pariah (unowned mongrels) dogs [19]. Such dogs risked reintroducing rabies into Britain or introducing it into other British colonies when their owners moved around the world [47,90]. Consequently, control measures included enforcing registration of owned dogs (purebred or pariah), levying a “dog tax” [19,86] and issuing badges or discs to be fitted to the collars of owned dogs [89]. Local authorities would thus be able to round up unowned dogs for destruction after 72 h, while straying owned dogs could be returned to their owners. Such a “tax and badge” policy was implemented in Shimla and Mussoorie and reported to function satisfactorily [91], while some local bodies were reportedly not keen on implementing these measures [92]. In military cantonments, kennels were set aside to isolate suspected rabid dogs for observation and destruction [93]. Some letters to newspapers proposed a ban on importation of dogs into India [49] while others argued that the quarantine of dogs imported into India would be pointless without first controlling rabies in the country [94].

As early as 1899, an annual administration report of the CVD in India highlighted the rise in number of rabies cases presented at Bombay and Lahore veterinary colleges and recommended that the GoI issue orders to prevent its spread in India [95]. However, whether rabies could ever be effectively controlled or eliminated in India appears to have been a contentious topic [96]. At the first meeting of veterinary officers in India, held at Lahore in 1919, veterinarians discussed the challenges of controlling stray dog populations and argued for mandatory licensing of all dogs and systematic destruction of strays [97]. The following resolution was adopted at this meeting—“That it is considered that any suitable measure that can be adopted for reducing and destroying the surplus population of dogs is desirable, but that it does not appear to be possible under the conditions prevailing in India to deal more effectively with the disease. Power should be given to veterinary practitioners to order the detention and destruction of dogs suffering from rabies.” Based on this resolution, the GoI appears to have advised local bodies to give veterinary officers relevant powers to perform these functions [84]. At the same time, it voiced doubts about the feasibility of implementing such control measures in rural areas and appears to have left it to municipalities and local self-governments to deal with the problem. This approach appears to have persisted throughout the period of British rule in India and there seem to have been no policies for nationwide rabies control in animals. It was also believed that rabies could not be eliminated in India as it was also maintained in wildlife [91].

After the world’s first rabies vaccine for dogs was developed in Japan in 1915, Umeno and Doi developed a single-dose canine rabies vaccine suitable for mass production in 1920 [98,99]. This vaccine was used for mass vaccination of dogs in Japan from 1924–25 [80,99]. In this context, experimental studies to develop a method of veterinary PEP (“anti-rabic inoculation of dogs bitten by rabid dogs”), presumably for valuable owned dogs, had begun at the Punjab Veterinary College in Lahore from 1915, also including horses in later years. These studies were inspired by work being conducted at the PI in Kasauli, fully recognising that the control of rabies in animals would benefit people as well [100]. A report of the Principal of this college in 1922–23 determined that rabies PEP of dogs could be considered an established mode of treatment (while also including caveats about conclusively establishing whether a dog was infected or not) [101]. Similar studies to develop preventive rabies vaccination in dogs were also started at the Madras Veterinary College from 1922 using rabies vaccine from the PI, Coonoor. Initial experiments were reported to be inconclusive because following vaccination, dogs from both experimental and control arms remained healthy after being infected with rabies virus [102].

Even prior to these studies, there are frequent reports of treatment of valuable animals exposed to rabies. For instance, two elephants owned by the Raja of Nilambur that were bitten by a rabid dog were given “anti-rabic treatment” in 1919–20, of which one elephant was confirmed by microscopic examination to have subsequently died of rabies. However the first record of the use of rabies vaccines for veterinary PEP, beyond those reported from the Punjab and Madras Veterinary Colleges, is found in a Madras CVD report from 1923–24, when ten animals were “treated with anti-rabic vaccine” at the veterinary hospital, Calicut (present-day Kozhikode in Kerala) [102].

From 1923, vaccines for veterinary use were issued from the PI, Coonoor, to veterinarians in Madras Presidency and Indian states [82]. The use of PEP to treat valuable animals (owned dogs, livestock at government livestock farms, equines and even a monkey) eventually started throughout most Indian provinces [102,103]. Vaccines were sourced from regional PIs, the Haffkine Institute in Bombay and the HK Mulford drug company in the USA [104]. By the 1930s and 1940s, veterinary PEP was being commonly administered at veterinary colleges and regional veterinary centres [45,73,104,105]. In response to rabies cases in Darjeeling municipality in 1933, legislation was enacted which required dog owners to vaccinate their dogs and to keep them muzzled or on a leash when in public [106]. A letter to a newspaper in 1935 complained that vaccination had failed to prevent the onset of rabies in some owned dogs [43]. Statistics of the number of patients treated at the PIs also reported figures for the number of animals vaccinated [68,107]. Between 1923 and 1948, 14,212 animals had been “prophylactically treated” at the PI in Coonoor [108]—these are all likely to have been owned.

However, the use of veterinary PEP seems to have been restricted to treating valuable owned animals, and mass vaccination of dogs for rabies control does not appear to have been seriously considered in British India. The studies conducted in Japan and the USA on preventive rabies vaccination of dogs as well as vaccination studies conducted at the Madras Veterinary College were discussed at the second meeting of veterinary officers in India, held at Calcutta in 1923. At this meeting, the opinion that rabies could never be eradicated in India persisted, and it was opined that vaccines would be useful only to reduce case numbers [84]. A newspaper report covering the conference stated that “The control of rabies in India constitutes one of the most difficult problems confronting both medical and veterinary authorities. The Conference resolved that the results of investigations upon the prophylactic vaccination of dogs against rabies should be referred to the Central Standing Advisory Committee on Epizootic Diseases and Research with a view to advising Government upon the desirability of enforcing measures of widespread inoculation of dogs against the disease.” [109]. We found no records to suggest that mass vaccination of dogs was ever considered by local authorities or the GoI. Mass culling of stray and rabid dogs and registration and, in later years, vaccination of owned dogs appear to have been the most widely implemented rabies control measures in colonial-era India. Such measures were even supported by Mahatma Gandhi, a central figure in the Indian struggle for independence, who stated that “The multiplication of dogs is unnecessary. A roving dog without an owner is a danger to society and a swarm of them is a menace to its very existence” [110].

## 5. Historic Animal Rabies Incidence in India

Annual administration reports of the CVD provide a detailed picture of the prevalence of animal diseases in British provinces in India. Provinces and presidencies were comprised of districts and municipalities with veterinary dispensaries and diagnostic laboratories, as well as veterinary colleges in some provinces. These institutions reported the number of cases of animal diseases treated or diagnosed. Infectious disease statistics from the earliest CVD reports (1887 onwards) focused solely on those affecting productive livestock (cattle, buffalo, sheep and goats) or equines such as rinderpest, foot-and-mouth disease, haemorrhagic septicaemia, anthrax, surra, strangles, etc. Rabies was mentioned only when it affected these species, and the disease was often included in the category “Other” diseases. It is only from 1903-04 onwards that rabies cases in livestock and dogs, most likely owned, were explicitly recorded in tables of disease summaries. Subsequently, the number of cases recorded increased significantly (Figure 1), which may have been partly driven by a number of provincial administrations framing rules for rabies prevention (e.g., Madras in 1923-24) [111].

Figure 1 presents the total number of animal rabies cases reported in all species (domestic and wildlife) between 1887-88 and 1950-51 across all provinces in British India. Case numbers reported from lower administrative levels (districts and municipalities), from regional veterinary colleges and/or diagnostic laboratories have been combined to present a breakdown by province/state in Figure 2. The Appendix A within contain more details on how rabies statistics were presented and compiled [112,113,114,115]. These reports indicate that animal rabies was endemic and widespread throughout all provinces in British India, affecting all species of domestic animals, most commonly dogs, and wildlife. Officials frequently highlighted their concerns about alarming increases in rabies cases and recommended implementation of control measures [96,100]. Outbreaks were occasionally reported, necessitating PEP treatment of several animals—for instance, the spike in cases in Bengal province in 1935-36 when 950 animal rabies cases were reported from all districts [46] (Figure 1). CVD staff were often exposed to rabies and had to undergo PEP at PIs [116].

CVD reports also highlight the wide variation in numbers of rabies cases reported from various provinces (Figure 2). In line with the lack of a consistent policy for rabies control in animals, there was little consistency in reporting of animal rabies cases [115]. Statistics compiled by Chakarabarti (2012) show that between 1880 and 1935 rabies caused an average of 160–170 human deaths per year in Punjab province [19]. However, the few animal rabies cases that are reported from Punjab province appear during the late 1800s and early 1900s, following which cases are reported only sporadically (Figure 2). It was often acknowledged that reported statistics of animal rabies incidence were likely to be underestimates of the true disease burden [115,117], which was sometimes attributed to a lack of public interest in reporting cases to veterinary officials [92]. There is a marked drop in the number of recorded animal rabies cases after 1941 (Figure 1), possibly because many provinces stopped reporting cases after this period (Figure 2).

Veterinary institutions charged a fee for admission and treatment of cases [118]. Some CVD reports indicate that while there was high demand for rabies PEP, the cost of treatment was unaffordable for poor animal owners and officials were unable to provide it free of cost [45,73]. A 1928–29 CVD report from Madras Presidency describes how poor dog and livestock owners could not afford the cost of PEP for their animals and were advised to administer “indigo blue” instead [114]. In later years, PIs charged for testing brain samples (ten rupees per brain sample in 1933–34), which made this service unaffordable for poor farmers [116]. Such barriers to treatment and diagnosis are likely to have influenced estimates of the true rabies burden and efforts to limit its spread.

## 6. The Origins of Its Neglect?

Chakrabarti (2012) discusses in detail the various ethical, moral and political debates around scientific research and treatment for rabies in colonial India [19]. It is debatable whether the motives behind the research and development of vaccines and control efforts targeting rabies and other diseases in India were purely altruistic or driven by imperial ambitions and a scientific fascination for tropical illnesses [77,119]. However, it is evident that colonial British governments in India invested time and resources to control rabies (among other diseases) in the country and so the disease could hardly be considered “neglected” from today’s perspective.

Given what has been described thus far, what might explain the neglect of rabies by public health practitioners and policymakers in modern-day India? Could it be driven by a perception that rabies had declined sufficiently to justify focusing on other more important human diseases? Dog bites and rabies clearly continued to be major public health concerns in India long after PIs started to save lives from the early 1900s. This is evident from newspaper reports and letters to newspapers from the public [35,39,40,42,93,120,121] as well as official reports and documents of the PIs and the GoI [63,77,122]. Annual reports from the PIs reveal a rapid rise in the number of patients vaccinated against rabies annually. The number of patients treated at the PI in Coonoor and its outcentres increased from under 200 in 1907 to 13,000 in 1935 [77], highlighting the high burden of animal bites in regions served by the institute. This increase was frequently attributed to wider awareness of the availability and benefits of “Pasteurian treatment” rather than any actual increase in rabies [61,63,123].

At the same time, it is not clear that human deaths from rabies reduced substantially in British India. As mentioned above, Chakarabarti (2012) found that around 160 to 170 people died of rabies every year in Punjab province between 1880 and 1935 [19]. This is despite the presence of the PI at Kasauli and the Lahore veterinary college in this province. In 1913, 243 rabies-related deaths were reported from the Central Provinces and Berar [63]. Similarly, 220 deaths were reported in India in 1922, with the report acknowledging that this figure was an underestimate of the true incidence [122]. A letter to the *Times of India* in 1911 speculated that the true number of dog bites and rabies deaths in India was likely to be much higher than those stated in reports from the PI [49]. It was recognised that not everyone completed the full course of post-exposure vaccinations and it was not possible to follow up on outcomes for all patients [122]. As human deaths from rabies in the general population outside those attending the PIs were not systematically recorded, the true death toll may have been much higher.

An examination of human disease statistics and discourses around public health in British India provides some hints about administrative and health priorities vis-à-vis rabies. Notwithstanding concerns raised by the public or officials of the CVD, provincial administrations did not consider rabies to be a concern compared to deaths from other contagious diseases or snakebites [19]. In the late 1800s, mortality statistics from British India included rabies deaths under the broad heading of “Injuries”, which covered a wide variety of conditions (suicide, wounds or accidents, snakebite, injuries caused by wild beasts, etc.) [124,125]. Before the establishment of the PI in India, rabies occasionally killed several soldiers stationed in India (69 deaths in 1879–80, 146 in 1885–86), though considerably fewer than the thousands of deaths every year due to diseases like cholera or smallpox [124,125].

With the advent of PIs and the race within scientific circles around the world to develop safer and more effective rabies vaccines, detailed records began to be maintained in India of the number of people vaccinated from broad ethnic (European and Indian) or religious (Muslim, Hindu, others) groups, those developing complications or dying post-vaccination, the number of patients that completed the full course of vaccination and differences in mortality rates between European and Indian patients [71,126,127]. Such statistics were published in annual reports of PIs [71], scientific journals [91] and, during the early 1900s, regularly included in annual reports presented to the UK Houses of Parliament [61,62,63,123,127,128,129,130]. These efforts served to establish the safety and efficacy of different vaccines being developed at the PIs and improve scientific understanding about rabies. For instance, in a letter written in 1911, W.F. Harvey, the then Director of the PI of Kasauli, recommended the collection of statistics by local bodies on the number of people bitten by rabid animals and who subsequently die without being vaccinated. His aim in suggesting this was to prove that the true mortality rate for rabies was much lower than that reported in statistics from Europe. He stated that this would involve “two or three years’ work only”, within which period he expected to prove his hypothesis [72]. In several significant respects, India was at the forefront of global research on rabies and the PI at Kasauli was central to this enterprise [19].

Statistical abstracts and reports of burden of illnesses in British India were split into sections—the first dealt with morbidity and mortality in the European Army in India, followed by the Native army (later referred to as the “Indian” army), the general population and jails [61,62,63,67,126,127,130]. Individuals treated for rabies at PIs were categorised as Europeans (including Eurasians/Anglo-Indians) and natives/Indians and further into soldiers and civilians. The number of Europeans vaccinated against rabies did not increase substantially over the years and few ever died of rabies. The number of Indians vaccinated increased annually, as did the number of recorded deaths (Table 1).

At the same time, overall rabies-related mortality continued to be much lower than mortality from other contagious diseases. Of 22,579 patients vaccinated between 1912 and 1916, only 135 (including 4 Europeans) died. This is in marked contrast with mortality from diseases like cholera (1,259,012 deaths between 1914 and 1917) and plague (1,599,088 deaths in the same period) [131]. At the second meeting of veterinary officers in 1923, it was even remarked that the money spent on rabies control in India would prove more beneficial if diverted for the control of cholera [132]. Indeed, diseases like cholera, plague, smallpox and malaria frequently caused extensive epidemics in India (e.g., the First Cholera Pandemic (1817–1821) [133], the plague epidemic in Bombay (1896) [50]) and high human mortality requiring active interventions by the state [50]. This focus on epidemic diseases would have been in marked contrast with rabies, which was, and continues to be to this day, characterised by fewer cases and only occasional outbreaks in animals [35,46]. Such outbreaks were handled by mass culling of dogs or jackals [87,89] and, following the development of vaccines, by PEP administration to human and animal bite victims.

As mentioned previously, there was also an emerging discourse around sanitation and urban improvement in colonial India from the late 1800s [50,51,52]. A range of sanitary reforms were implemented from this period, particularly aimed at improving the health of European army personnel who initially suffered significantly higher morbidity and mortality from epidemics in India when compared to Indian soldiers. Sanitary measures such as the provision of piped and filtered water, relocating barracks from swampy areas and improvements in drainage and preventive vaccination against smallpox and plague caused a remarkable and consistent decline in morbidity and mortality among British troops in India [50]. Such sanitary measures would have had little impact on rabies, which would not have been seen to be as amenable to human modification of environmental conditions. Preventive vaccination of humans against rabies, as practised for smallpox, would hardly have been considered necessary, given the sporadic nature of the disease. These epidemiological characteristics of rabies are likely to have greatly influenced colonial perceptions of what diseases could be reasonably controlled through public health interventions.

A similar situation existed with rabies in animals in India. CVD officials were more concerned with the treatment, control and prevention of diseases affecting equines and livestock, which were largely unaffected by rabies. For instance, in 1935-36, when Bengal province experienced rabies outbreaks in multiple districts and recorded 950 cases (Figure 2), the number of livestock deaths from rinderpest and haemorrhagic septicaemia was 35,281 and 3989, respectively [46]. This is unlike the situation reported in Trinidad, for instance, where between 1925 and 1958 repeated outbreaks of rabies transmitted by bats threatened the livestock industry, prompting widespread vaccination campaigns for cattle and efforts to destroy bat populations. As a result, rabies was accorded high government priority for control and elimination in Trinidad with WHO assistance, and much research was conducted on this topic [136]. Dog rabies had also been eliminated in Britain in 1902, and barring occasional outbreaks seeded by dogs brought into the country, rabies ceased to be the significant domestic public health concern it once had been for British politicians and policymakers [47]. This may also have contributed to the gradual loss of interest in investing in rabies control and prevention in British India.

The success of Semple’s vaccine in reducing human rabies deaths in India was soon recognised by the global scientific community, and it began to be widely used around the world [81]. By the 1930s, statistics from the PIs continued to be published in scientific journals [108,134,135] but were no longer included in reports to the UK Parliament. During this period, one also finds a return to the practice of including rabies deaths under the head of “Injuries” [137,138]. While research to make the Semple vaccine safer did continue, the key personnel driving this research left the PI or were transferred, and research on other diseases began to take precedence. The PI at Kasauli was shut down in 1939, with work being shifted to the Central Research Institute next door [19]. Research on rabies vaccines and diagnosis continued to be conducted at the Coonoor PI after Indian independence in 1947, spearheaded by the institute’s Director N. Veeraraghavan [139], but rabies no longer appears to have been accorded the same priority it once was in British India during the early 20th century.

Thus, despite the importance given to rabies in India with the advent of Pasteurism, a combination of factors is likely to have contributed to its eventual decline in administrative and public health priorities. In particular, the success of Semple’s vaccines in preventing human rabies deaths will have influenced administrative officials to prioritise scarce resources towards competing and more pressing public health interventions (e.g., improving sanitation and addressing epidemics). A point to this effect is made in an anecdote in a CVD report about rabies control becoming a priority only “when the deaths amongst humans numbered some scores annually, and a genuine feeling of alarm for personal safety was felt” [140]. In this respect, rabies in British India may have become a victim of its own success, something which is recognised today in Latin America as canine rabies control has become more effective and human mortality has reduced dramatically [141].

## 7. Impacts on Present-Day Debates in India

Much literature exists on the medical history of a range of infectious diseases that caused major epidemics in British India, including malaria, cholera, plague and smallpox [50,51,142]. This review has examined the historical treatment in British India of rabies as a public health concern, a topic which has received relatively less attention. From being a widespread and untreatable illness, rabies rose to become a driver of Pasteurism in India and globally and a key component of British colonial scientific enterprise. The disease, however, eventually declined in administrative and public health priorities in India towards the end of colonial rule—a decline that has yet to be reversed in modern-day India. In charting this history of rabies, the review highlights features of the colonial administrative, scientific and societal approach to dealing with this disease in India that remarkably persist in the country nearly a century later.

Key among these are the interrelated issues of an absence of a rabies control policy at the national level and of systems for rabies surveillance in humans and animals [7]. Notwithstanding the existence of a National Rabies Control Programme [14], India lacks a well-considered roadmap with realistic milestones to chart progress towards effective national rabies control, let alone elimination by 2030, the target set by the WHO for global elimination of human deaths from dog-mediated rabies [4]. Policy formulation and implementation continue to be the responsibility of states and local bodies, and these are consequently inconsistent. For instance, only two states (Tamil Nadu and Sikkim) have made human rabies notifiable [143]. As was the case in colonial India, animal rabies is not seen as an economically relevant disease affecting animal production systems and hence is not prioritised for control by agriculture or animal husbandry ministries [144].

Rabies in animals was widespread in space and time across British India (Figure 2). The mean number of animal rabies cases recorded between 1903–04 and 1950–51 was 522. This is likely to be an underestimate given that reporting was unsystematic and not mandatory and reported numbers do not include cases from most princely states and territories not under direct British control. The human population of India has risen from 361 million in 1951 to over 1.2 billion in 2011 [145], and the population of dogs has increased correspondingly. In the absence of any comprehensive rabies control measures, it therefore stands to reason that the number of animal rabies cases will also have increased significantly. Although animal rabies is notifiable in India today, disease reporting is acknowledged to be unreliable even by rabies experts in India [143], and rabies statistics such as those reported by the CVD are difficult to access. This makes the task of estimating the true prevalence in animals extremely difficult. Such gaps in knowledge of the human and animal disease burden and patchy awareness of rabies as a public health threat [146], even among medical health professionals [147], significantly hinder the development of political, scientific and societal urgency to address this burden, particularly in rural areas which bear the biggest brunt [5].

One positive change from the colonial-era approach to rabies control in India is with respect to dog population management (DPM). Although culling of dogs continued to be the mainstay of DPM and rabies control efforts for decades after independence, the Animal Birth Control (ABC) (Dogs) Rules established in 2001 outlawed this inhumane measure [148]. It was replaced by a policy of sterilisation and anti-rabies vaccination (ABC-ARV), carried out once during a dog’s lifetime, after which it was returned to its original location. However, in the absence of scientifically robust methods to obtain reliable estimates of dog population sizes, ABC-ARV is implemented in a haphazard and uncoordinated manner across local bodies and states, involving various public [149,150] and private entities [151]. The policy also does not account for the need to revaccinate sterilised dogs to maintain anti-rabies immunity in the dog population and, consequently, the possibility that these dogs may continue to bite and transmit rabies to other dogs and people [152]. Rule 10 of the ABC Rules prohibits the euthanasia of dogs suspected to be rabid, instead requiring such dogs to be isolated until they die naturally of rabies [148], followed by laboratory confirmation of disease [153]. This is clearly a welfare issue for infected dogs. Enforcement of the ABC rules is also inconsistent, and culling of dogs still occurs occasionally throughout India [154], often in retaliation to incidents of injuries or deaths from dog attacks [155].

There is also a flawed perception of ABC-ARV as more than just a DPM tool. Consequently, this measure is increasingly viewed as the primary rabies control measure in India, perceived to be unsuccessful in reducing disease only because of ineffective and/or inadequate implementation [156]. This perception is despite the fact that the WHO itself recommends sterilisation of dogs only as a supportive measure to maintain levels of rabies vaccination coverage achieved through mass rabies vaccination, accepted as the most scientific method for rabies control [4]. A recent study was unable to evaluate the role of surgical sterilisation in controlling dog rabies due to poor data collection or reporting and recommended that mass vaccination should continue to be the control method of choice [157]. The ABC-ARV policy also finds support through discourses that argue for the continued existence of street dogs as “integral inhabitants of the multispecies city” [156].

Wang (2019) describes the conflict that existed in New York City from the early 20th century, between the American Society for Prevention of Cruelty to Animals and the Department of Health, over population control and muzzling of the city’s free-roaming dogs [158]. No such conflict appears to have existed over dog culling in colonial India. On the contrary, there is in India today an impasse, along the lines of that which existed in New York, between two conflicting perspectives of the place of dogs on the streets, with direct impacts on rabies control efforts. On the one hand is the view that there should be a holistic approach to control stray dog populations on public health, wildlife conservation [159] and animal welfare grounds. This approach would require enforcing responsible dog ownership, civic waste and humane dog population management and a national mass rabies vaccination programme [4,152], eventually leading to the elimination of free-roaming dogs. On the other hand is the view, held primarily by animal welfare campaigners, that dogs have the right to exist on the streets and to be fed by people [160]. This latter view consequently favours ABC-ARV as the most appropriate DPM and rabies control measure, notwithstanding its drawbacks. Nadal (2020) discusses various aspects of the conflicting perspectives about DPM and rabies control, as well as the complex social, cultural and political contexts within which people and dogs interact in two major cities in North India [161].

In this respect, the ABC-ARV policy has made it difficult to adopt comprehensive measures to deal with the persistent threat of rabies posed by the large populations of unowned free-roaming dogs in India [144], particularly implementation of mass dog vaccination. Despite evidence from the 1920s that mass vaccination of dogs successfully reduces rabies incidence and can eliminate it [98,99], there were no attempts to implement such a measure in colonial India. Instead, it was widely considered that rabies could never be effectively eliminated in India. This perception continues to hold sway at the highest levels of government to this day with the view that logistical constraints make mass vaccination of dogs unfeasible in India [162]. It is left to state administrations to implement mass vaccination policies, commonly in partnership with nongovernmental organisations [163,164].

In another unfortunate parallel with the colonial era, there is little emphasis on promoting responsible dog ownership practices such as confinement and vaccination of owned dogs in India. It is unclear how successful attempts were by colonial administrations to enforce registration and identification of owned dogs. While several local bodies have now made such measures mandatory in India, they are poorly enforced [165]. Rabies vaccination, while largely confined to owned dogs, primarily valuable pure breeds, is also not mandatory. This often results in poor compliance with vaccination regimens [165], especially in the case of owned mongrel dogs (i.e., those that do not belong to any specific breed), from poorer households who have greater difficulty accessing veterinary services. A high proportion of this latter category of dogs are also poorly confined, frequently resulting in the birth of unwanted pups and increased risk of contracting rabies from interactions with free-roaming unowned dogs [166].

## 8. Conclusions

Notwithstanding poor availability of disease data, the case may be made that rabies does not impose the kind of human health burden in India that diseases such as tuberculosis, malaria and HIV do. Consequently, rabies control may not be seen as a cost-effective public health investment, a view that was certainly shared by public health practitioners in British India [132]. Such a perspective, however, fails to consider the impact of rabies on individuals from rural backgrounds, particularly children [8], and the near certainty of death in the absence of access to treatment before symptoms appear. As an entirely vaccine-preventable disease disproportionately affecting the poor in low- and middle-income countries, preventing unnecessary human rabies deaths and suffering by addressing barriers to access to human PEP is an important means of achieving social justice [167]. At the same time, the cost of human PEP provision can be substantial (30 million US dollars over an unspecified timeframe in India, by one estimate) [143]. Rabies control through mass dog vaccination has been consistently shown to be more cost-effective in preventing human rabies deaths [168,169]. With the science and tools for rabies control already existing, rabies elimination is low-hanging fruit and a textbook example of the One Health approach in action. This is well recognised even in India, with zoonotic disease prioritisation exercises frequently identifying rabies as one of the main diseases for targeting control efforts [170,171]. Political will has been key to implementing effective control measures in many countries around the world [4] and is the primary factor currently hindering progress on this front in India today.

Rabies in British India was clearly not a “neglected” public health concern. Early rabies vaccines were highly effective in saving human lives, although there remained a poor understanding of the true disease burden in Indian society. These factors, combined with changing priorities of colonial British governments, in all likelihood contributed to a progressive loss of priority of rabies control in the face of the vast array of competing infectious disease and public health challenges in British India. It may be possible to argue that the current neglect of rabies in India is a legacy, albeit unintended, of British colonial rule, but this clearly is no justification for carrying on in the same vein. Current public health professionals and policy makers should look to the extensive historic and current scientific literature on evidence-based rabies control measures to formulate a strategy to achieve the lofty goal of elimination of dog-mediated human rabies deaths by 2030. Equally crucial will be measures to deal with the extensive free-roaming dog population in India, without which rabies control efforts will become unsustainable in the long run.

## Figures and Tables

**Figure 1 tropicalmed-05-00162-f001:**
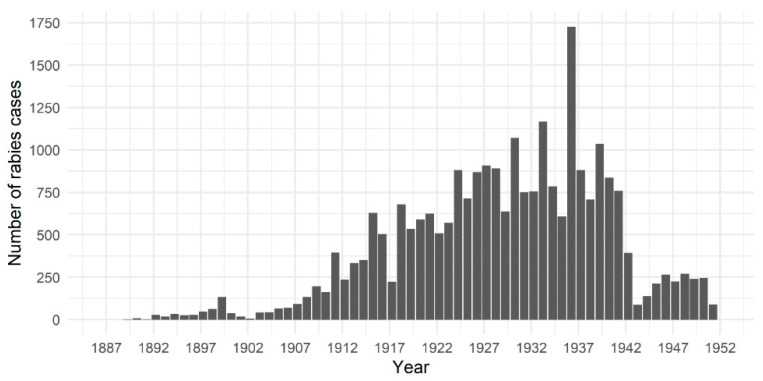
Total number of rabies cases reported each year in all animal species in British India between 1887 and 1951 (annual data span April to the following March, e.g., April 1887 to March 1888). Statistics were compiled from annual reports of the Civil Veterinary Department of the colonial British Government of India, available at https://digital.nls.uk/indiapapers/.

**Figure 2 tropicalmed-05-00162-f002:**
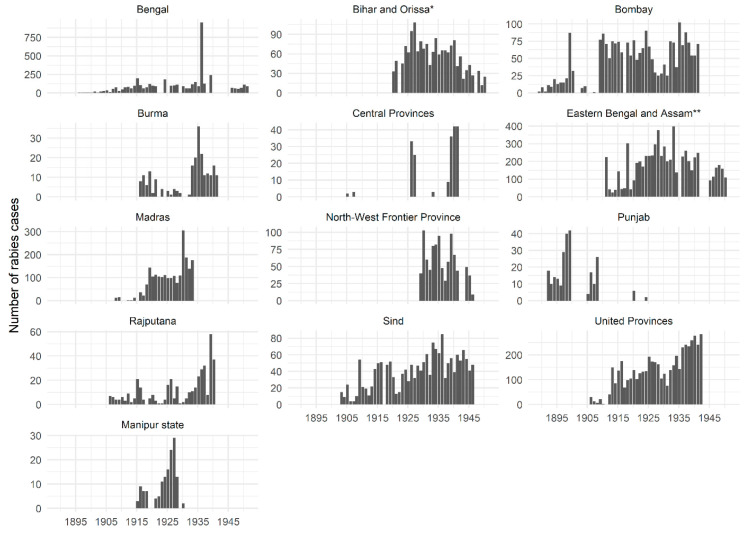
Annual rabies incidence in all animal species between 1887 and 1951 (annual data span April to the following March, e.g., April 1887 to March 1888) in every British province in India (except Baluchistan for which no data were available) and the princely state of Manipur. Statistics were compiled from annual reports of the Civil Veterinary Department of the colonial British Government of India, available at https://digital.nls.uk/indiapapers/. Note the different scale of the *y*-axis for each region.

**Table 1 tropicalmed-05-00162-t001:** Number of people given rabies post-exposure prophylaxis at various Pasteur Institutes in India. Category totals may not always match as the breakdown of the number of patients treated and the number of deaths was not always explicitly reported.

Year	Numbers Treated (Number of Deaths)			Reference
	European	Native/Indian	Total	
1900–1901 ^1^	146 (1)	175 (9)	321 (10)	[130]
1901–1902 ^1^	215 (2)	328 (11)	543 (13)	[130]
1902–1903 ^1^	269 (1)	315 (12)	584 (12)	[130]
1903–1904 ^1^	248 (0)	364 (10)	612 (10)	[130]
1904–1905 ^1^	307 (0)	570 (12)	877 (12)	[130]
1905–1906 ^1^	342 (2)	803 (19)	1145 (21)	[130]
1906–1907 ^1^	452 (2)	846 (17)	1308 (19)	[130]
Interim, 09/08–31/12, 1907 ^1^	146 (1)	373 (4)	519 (5)	[130]
1908 ^2^	342 (2)	1047 (24)	1729 (26)	[128,130]
1909 ^2^	675 (3)	1920 (25)	2595 (28)	[126,130]
1910 ^2^	575 (0)	2325 (43)	2900 (43)	[127,130]
1911 ^2^	297(1)	2911 (50)	3208 (51)	[130]
1912 ^2^	400 (0)	4388 (59)	4788 (59)	[61]
1913 ^2^	2 (2)	5271 (66)	5273 (68)	[129]
1914 ^2^	NA (1)	NA (60)	5795 (61)	[63]
1915 ^2^	468 (1)	6409 (41)	6877 (42)	[123]
1933 ^1^	1356 (0)	14,582 (83)	15,938 (83)	[134]
1936 ^1^	1357 (0)	NA (97)	NA (97)	[135]
1938 ^3^	NA (NA)	NA (NA)	12,396 (21)	[68]

^1^ Figures for Kasauli institute only; ^2^ figures combined for all Pasteur institutes in India, where available; ^3^ Coonoor institute and its subsidiary centres only; NA—not available.

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
