# Peer review of "Rabies as a Public Health Concern in India—A Historical Perspective"

_tropicalmed, 2020, doi:10.3390/tropicalmed5040162_

Round 1
Reviewer 1 Report
The review article by Radhakrishnan et al. provides a historical perspective on Rabies as a public health concern in India. As can be expected from a good and comprehensive review, the authors have used a plethora of sources dating back to colonial times. The depth of information and the relevance to the topic had not been dealt in detail in previous articles (e.g. Tarantola et al., 2019) nor in book chapters (e.g. RABIES - Scientific Basis of the Disease and Its Management, 4th edition, Chapter 1: A history of rabies—The foundation for global canine rabies elimination).
In the current momentum to eliminate human rabies of dog bite origin, it becomes increasingly clear that socio-cultural components are an eminent factor. To this end, the authors provide valuable background information that can partly explain the current situation as regards rabies control in India.
I only have marginal comments to the authors that they may use to improve the manuscript.
Ll 71: “A” should not be capitalized
Ll 205: Laboratory should not be capitalized
Author Response
Sreejith Radhakrishnan
Department of Infectious Disease Epidemiology
Imperial College London
St. Mary’s campus, Norfolk place
London W2 1PG, UK
09.10.2020
Dear editors,
Thank you for considering our manuscript. Please find detailed responses to each of the reviewer’s comments below.
Yours sincerely
Sreejith Radhakrishnan
Response to Reviewer 1 comments
Thank you for your encouraging feedback and comments. Please find our responses to specific comments below:
- Line 71: “A” should not be capitalized.
Response: this has been corrected.
- Line 205: Laboratory should not be capitalized
Response: this has been corrected.
Additional changes made:
The following changes/ additions have also been made to the manuscript, in response to feedback to the preprint of the manuscript from various sources:
- Line 118 – has been edited as follows:
‘We used the search terms ‘rabies’ (and variations of its spelling – ‘rabbies’, ‘rabeis’ and ‘rabes’), ‘hydrophobia’ and ‘India’ to review a range of historical archives, online and physical documents …..’
- Line 374 – the following sentence has been added.
‘Such measures were even supported by Mahatma Gandhi, a central figure in the Indian struggle for independence, who stated that ‘The multiplication of dogs is unnecessary. A roving dog without an owner is a danger to society and a swarm of them is a menace to its very existence’ [ref. 110].’
- Line 618 – the following sentence has been added.
‘Enforcement of the ABC rules is also inconsistent and culling of dogs still occurs occasionally throughout India [ref. 154], often in retaliation to incidents of injuries or deaths from dog attacks [ref. 155]’.
- Line 643 – the following sentence has been added.
‘Nadal (2020) discusses various aspects of the conflicting perspectives about DPM and rabies control, as well as the complex social, cultural and political contexts within which people and dogs interact in two major cities in North India [ref. 161].’
- Line 696 – the following sentence has been added.
‘Equally crucial will be measures to deal with the extensive free-roaming dog population in India, without which rabies control efforts will become unsustainable in the long run.’
Four new references (nos. 110, 154, 155 and 161) have been added.
Reviewer 2 Report
The authors should be congratulated on this thorough, balanced and eloquent account of the history of rabies in British India, which sheds light on modern-day governmental perspective to rabies control in India and the possible reasons for the lack of prioritization of this issue. This review will be of interest to the readership of Tropical Medicine and Infectious Diseases and has the potential to generate helpful discussion on how historic inaction to control rabies at its source may be addressed at the level of government. I have very minor specific comments and otherwise recommend this article for publication.
Specific comments
Some type errors for correction (e.g. line71).
Line 202 – for the reader’s interpretation it would be useful to report what this amount would be in modern-day equivalent cost after accounting for inflation.
Figures 1 and 2. Captions would benefit from clarification. It is not immediately clear what is meant by “…between 1887 -1888 and 1950-51…”, however it seems that the annual data was reported spanning calendar years. Perhaps it would be clearer to describe something like “Total number of cases … between 1887 and 1951 (annual data spanned April to the following March).”
The formatting of several references in the Reference section require review. For example References 88, 92, 100, 101, 102, 103, 105, 113 – 117, 131, 139, do not state the journal name etc. Reference 91 doi link is not functional
Author Response
Sreejith Radhakrishnan
Department of Infectious Disease Epidemiology
Imperial College London
St. Mary’s campus, Norfolk place
London W2 1PG, UK
09.10.2020
Dear editors,
Thank you for considering our manuscript. Please find detailed responses to each of the reviewer’s comments below.
Yours sincerely
Sreejith Radhakrishnan
Response to Reviewer 2 comments
Thank you for your encouraging feedback and comments. Please find our responses to specific comments below:
- Some type errors for correction (e.g. line71).
Response: The manuscript has been edited to correct typos and errors in spelling and capitalization.
- Line 202 – for the reader’s interpretation it would be useful to report what this amount would be in modern-day equivalent cost after accounting for inflation.
Response: The inflation-adjusted equivalent cost in 2019 of £100 in 1911 (the year of publication of the reference cited) was calculated to be £11,867.71, using the Inflation Calculator from the Bank of England’s website (https://www.bankofengland.co.uk/monetary-policy/inflation/inflation-calculator).
Lines 201-202 have therefore been edited to read as follows:
‘These journeys were a major financial burden for the GoI, by one estimate costing £100 per person treated in 1911 (approximately £12,000 per person in 2019 terms) [58].’
- Figures 1 and 2. Captions would benefit from clarification. It is not immediately clear what is meant by “…between 1887 -1888 and 1950-51…”, however it seems that the annual data was reported spanning calendar years. Perhaps it would be clearer to describe something like “Total number of cases … between 1887 and 1951 (annual data spanned April to the following March).”
Response: The captions for figures 1 and 2 have been edited as suggested, to read as follows:
Figure 1. ‘Total number of rabies cases reported each year in all animal species in British India between 1887 and 1951 (annual data spanned April to the following March e.g. April 1887 – March 1888 and so on).’
Figure 2. ‘Annual rabies incidence in all animal species between 1887 and 1951 (annual data spanned April to the following March e.g. April 1887 – March 1888 and so on)…’
- The formatting of several references in the Reference section require review. For example References 88, 92, 100, 101, 102, 103, 105, 113 – 117, 131, 139, do not state the journal name etc. Reference 91 doi link is not functional
Response: The formatting of all references cited in the paper has been reviewed and corrected to ensure that details such as journal name and webpage title are displayed correctly. We have verified that the DOI link for reference 91 redirects users to the correct webpage of The Lancet.
Additional changes made:
The following changes/ additions have also been made to the manuscript, in response to feedback to the preprint of the manuscript from various sources:
- Line 118 – has been edited as follows:
‘We used the search terms ‘rabies’ (and variations of its spelling – ‘rabbies’, ‘rabeis’ and ‘rabes’), ‘hydrophobia’ and ‘India’ to review a range of historical archives, online and physical documents …..’
- Line 374 – the following sentence has been added.
‘Such measures were even supported by Mahatma Gandhi, a central figure in the Indian struggle for independence, who stated that ‘The multiplication of dogs is unnecessary. A roving dog without an owner is a danger to society and a swarm of them is a menace to its very existence’ [ref. 110].’
- Line 618 – the following sentence has been added.
‘Enforcement of the ABC rules is also inconsistent and culling of dogs still occurs occasionally throughout India [ref. 154], often in retaliation to incidents of injuries or deaths from dog attacks [ref. 155]’.
- Line 643 – the following sentence has been added.
‘Nadal (2020) discusses various aspects of the conflicting perspectives about DPM and rabies control, as well as the complex social, cultural and political contexts within which people and dogs interact in two major cities in North India [ref. 161].’
- Line 696 – the following sentence has been added.
‘Equally crucial will be measures to deal with the extensive free-roaming dog population in India, without which rabies control efforts will become unsustainable in the long run.’
Four new references (nos. 110, 154, 155 and 161) have been added.
Reviewer 3 Report
This is well written review paper describing the historical aspects of rabies treatment both in humans and animals in British India. The paper provides excellent background to help to design roadmap with realistic milestones to chart progress towards effective national rabies control program to achieve in India elimination of human deaths from dog-mediated by 2030, the target set by the WHO.
Author Response
Sreejith Radhakrishnan
Department of Infectious Disease Epidemiology
Imperial College London
St. Mary’s campus, Norfolk place
London W2 1PG, UK
08.10.2020
Dear editors,
Thank you for considering our manuscript. Please find detailed responses to each of the reviewer’s comments below.
Yours sincerely
Sreejith Radhakrishnan
Response to Reviewer 3 comments
Thank you very much for your encouraging feedback.
Additional changes made:
The following changes/ additions have also been made to the manuscript, in response to feedback to the preprint of the manuscript from various sources:
- Line 118 – has been edited as follows:
‘We used the search terms ‘rabies’ (and variations of its spelling – ‘rabbies’, ‘rabeis’ and ‘rabes’), ‘hydrophobia’ and ‘India’ to review a range of historical archives, online and physical documents …..’
- Line 374 – the following sentence has been added.
‘Such measures were even supported by Mahatma Gandhi, a central figure in the Indian struggle for independence, who stated that ‘The multiplication of dogs is unnecessary. A roving dog without an owner is a danger to society and a swarm of them is a menace to its very existence’ [ref. 110].’
- Line 618 – the following sentence has been added.
‘Enforcement of the ABC rules is also inconsistent and culling of dogs still occurs occasionally throughout India [ref. 154], often in retaliation to incidents of injuries or deaths from dog attacks [ref. 155]’.
- Line 643 – the following sentence has been added.
‘Nadal (2020) discusses various aspects of the conflicting perspectives about DPM and rabies control, as well as the complex social, cultural and political contexts within which people and dogs interact in two major cities in North India [ref. 161].’
- Line 696 – the following sentence has been added.
‘Equally crucial will be measures to deal with the extensive free-roaming dog population in India, without which rabies control efforts will become unsustainable in the long run.’
Four new references (nos. 110, 154, 155 and 161) have been added.